# Imaging Review of Pediatric Benign Osteocytic Tumors and Latest Updates on Management

**DOI:** 10.3390/jcm10132823

**Published:** 2021-06-26

**Authors:** Jignesh Shah, Darshan Gandhi, Ankita Chauhan, Saurabh Gupta

**Affiliations:** 1Department of Pediatric Radiology, University of Tennessee Health Science Center, Memphis, TN 38103, USA; achauha3@uthsc.edu; 2Department of Radiology, Northwestern University Feinberg School of Medicine, Chicago, IL 60611, USA; darshan.gandhi@northwestern.edu; 3Department of Pediatric Radiology, SUNY Upstate Medical University, Syracuse, NY 13210, USA; guptasau@upstate.edu

**Keywords:** pediatric benign osteocytic tumor, enostosis, osteoma, osteoid osteoma, osteoblastoma, radiofrequency ablation

## Abstract

Pediatric benign osteocytic tumors include osteoma, enostosis, osteoid osteoma, and osteoblastoma. In pediatric populations, benign bone tumors are more common than malignancies. Benign osteocytic tumors may have a unique clinical presentation that helps narrow the differential diagnosis. A systemic imaging approach should be utilized to reach the diagnosis and guide clinicians in management. Radiographs are the most prevalent and cost-effective imaging modality. Cross-sectional imaging can be utilized for tissue characterization and for evaluation of lesions involving complex anatomical areas such as the pelvis and spine. Computed Tomography (CT) is the modality of choice for diagnosis of osteoid osteoma. CT scan can also be utilized to guide radiofrequency ablation, which has been found to be highly effective in treating osteoid osteoma and osteoblastoma. Enostosis is a no-touch lesion. Osteoma is commonly located in the paranasal sinuses. Osteoma needs an excision if it causes complications due to a mass effect.

## 1. Introduction

Pediatric benign osteocytic tumors include osteoma, enostosis (bone island), osteoid osteoma, and osteoblastoma. Benign bone tumors occur more frequently than malignancies in the pediatric population [1]. Benign osteocytic tumors may have a unique clinical presentation that helps narrow the differential diagnosis. Clinicians should be familiar with the clinical and imaging presentation of these benign bone tumors so that they can assure the patient and their family about the benign nature of the lesion, thus avoiding anxiety and preventing unnecessary imaging or intervention.

## 2. Clinical Evaluation and Imaging Approach

Most commonly, osteocytic tumors are detected incidentally on imaging studies performed for some unrelated, nonspecific symptoms. One should always consider the possibility of a benign bone tumor in a child with unexplained musculoskeletal pain. An osteoid osteoma may present as localized pain due to the presence of intralesional prostaglandins. Osteoblastoma of the spine may present as lower extremity pain, mimicking a disk pathology, or as painful scoliosis; in either case, the condition may remain unnoticed until appropriate imaging of the spine reveals the tumor. Based on the classification by the Musculoskeletal Tumor Society, benign tumors of the bone are staged into three stages, according to their radiographic appearance and clinical presentation [2]. Stage one lesions are static or latent lesions, typically self-resolving (no surgical intervention required). Stage two lesions are active but stay within the confines of the bone and are associated with bone destruction or remodeling (may need intervention if they result in structural fragility or are markedly symptomatic). Stage three lesions are active and regionally aggressive and extend outside the cortex into surrounding soft tissues, and require surgical treatment.

Radiographs are the most prevalent first-line imaging modality to evaluate pediatric benign bone tumors. Radiographs help assess the biologic behavior and probable histologic diagnosis. Cross sectional imaging is usually performed to evaluate the extent of the lesion and staging [3].

## 3. Osteoma

Osteoma, a benign, slowly growing neoplasm of compact and trabecular bone in varying proportions, is most often found incidentally. Males are more frequently affected than females [4]. It typically occurs between the third and fifth decades. Osteoma is typically seen in bones developed via membranous ossification (for example, skull and facial bones) [5]. It is usually solitary and can be sessile or pedunculated. It primarily arises from the outer table but rarely can arise from the inner table (intradiploic osteoma) [6,7]. Unlike ossified meningiomas, inner table osteomas neither enhance nor have a soft tissue component [6]. Gardner syndrome has a known association with multiple osteomas [8].

Osteomas are mostly juxtacortical lesions (arising from the periosteum) involving the paranasal sinuses (mainly frontal and ethmoidal), skull, mandible, and maxilla [9]. The most common site of involvement in a cranial osteoma is the frontal sinus [10]. Osteomas are usually asymptomatic. Based on their location, they may cause headaches, sinusitis, and other symptoms related to obstruction of the paranasal airspace.

A small circumscribed bony lump on the skull vault is called a button osteoma [11]. The differential diagnosis of a button osteoma includes metastatic cancer, subcutaneous hematoma, lipoma, epidermal cyst, and ballooned osteoma.

Pathologically, three different histologic patterns have been described:Ivory osteoma (also known as eburnated osteoma) contains dense bone without a haversian system.Mature osteoma (also known as osteoma spongiosum) is histologically similar to normal bone. Mature osteoma is composed of trabecular bone, often with marrow.Mixed osteoma contains features of both ivory and mature osteoma.

On all imaging modalities, an osteoma is a sharply defined bony surface lesion arising from the cortex. On CT, osteoma appears as a juxtacortical, circumscribed, homogeneous, and sclerotic lesion (Figure 1). MRI demonstrates a homogenous hypointense T1 signal but has a variable T2 appearance (depending on the amount of compact and trabecular bone) [6].

Osteoma does not require surgical treatment unless the location or size of the lesion causes a mass effect upon the adjacent structures (orbit, sinus, brain). Osteomas can be managed by direct excision or endoscopic resection. Contemporary practice prefers endoscopic resection for cosmetic reasons (decreased facial scarring), and its reduction in neural complications by preserving cutaneous nerves [12]. Surgical removal of button osteoma (including ostectomy, curettage, endoscopic surgery, and CO2 laser cauterization) is usually performed for improved cosmesis. Asymptomatic small frontal osteomas may be followed up at 6-month intervals by physical examination [11].

## 4. Enostosis (Bone Island)

Enostosis, also known as bone island, results from a developmental abnormality during enchondral ossification, causing dense and compact cortical bone focus within the spongiosa or cancellous bone [13]. It is an incidental, mostly asymptomatic, and benign finding that is no longer on the World Health Organization’s histologic classification of bone tumors [8,13]. There is no gender or age predilection, and most measure from 1 mm to 2 cm in diameter. An enostosis more than 2 cm in diameter is known as a giant bone island [8]. A large bone island may cause pain.

Bone islands mainly involve the axial skeleton (spine, pelvis, and ribs) and long bones such as the femur (epiphysis or metaphysis) [8]. On radiographs and CT, enostosis appears as an oval, homogeneous, intramedullary sclerotic focus with a density similar to the cortex [8]. It frequently has a spiculated or thorny margin that blends with the adjacent trabecula or cortex (Figure 2) [14,15]. No bone destruction, periosteal reaction, or soft-tissue involvement is demonstrated [16]. Under MRI and on all the pulse sequences it appears as a low signal focus within an otherwise normal marrow signal. Most enostoses show radiotracer accumulation similar to background bone on 99mTc–methylene diphosphonate bone scans [14]. When an enostosis shows hot uptake under bone scintigraphy or PET/CT, it may be confused with a malignant lesion (metastasis) [16]. With time, a bone island may enlarge in size. A biopsy is warranted if it shows a growth of more than 25% over six months or 50% in one year [13,17].

Osteopoikilosis (osteopathia disseminata), a heritable disorder, is characterized by multiple small rounded or ovoid bone islands, mostly involving the long bones (both sides of the joints), tarsals, or carpals (Figure 3) [18].

The differential diagnosis includes multiple sclerotic bone lesions seen in tuberous sclerosis complexes and osteoblastic bone metastases, such as in the setting of multifocal osteosarcoma. A metastatic lesion usually shows increased activity on bone scintigraphy and demonstrates a “halo sign” on fluid-sensitive MRI sequences (a halo of increased T2 signal in the surrounding marrow). When present, a halo sign should prompt biopsy of the sclerotic lesion as it is indicative of an active tumoral lesion and is a highly specific (about 99%) feature of metastasis [14]. Planar bone scintigraphy, when used alone, should not be used to exclude or warrant a biopsy of any individual sclerotic lesion. SPECT can better detect individual sclerotic metastasis compared to planar bone scintigraphy. 18F-NaF-PET/CT shows an improved ability to detect sclerotic bone metastases compared to bone scintigraphy, with sensitivities above 90% [14]. According to Ulano et al., a mean CT attenuation threshold of 885 HU and a maximum attenuation threshold of 1060 HU can differentiate untreated osteoblastic metastasis from enostosis, with 95% sensitivity and 96% specificity [13]. With Radiomics, a technique that extracts and analyzes quantitative information to maximize information from clinical images without further acquisition of the image, quantitative analysis of tumoral heterogeneity is made possible. A radiomics model can assist an inexperienced radiologist in differentiating bone islands from osteoblastic metastasis while analyzing an incidental sclerotic lesion on CT [16].

Enostoses are “do not touch” lesions that do not warrant treatment.

## 5. Osteoid Osteoma

Osteoid osteoma (OO), a painful benign bone lesion, is more common in males (M: F = 1.6 − 4:1) and primarily affects patients between 5 and 25 years of age [8,18,19]. It comprises 10–12% of all benign bone tumors and 2–3% of all primary bone tumors [18,19]. It has a classical clinical presentation of worsening night pain that is relieved with a nonsteroidal anti-inflammatory drug. Long bones are most affected (65–80%) with the femur being the most common. Approximately 66% of femoral lesions are located in the intertrochanteric region. The nidus is usually less than 1.5–2 cm in diameter [8]. Depending on the location of OO within a bone (cortical (most common; 75%), medullary (about 20%), and subperiosteal (<5%)), the imaging appearance varies [19]. A typical cortical lesion presents as a less than 2 cm cortically-based lucency in the diaphysis, surrounded by extensive fusiform sclerosis. In comparison, there is less sclerosis surrounding the epiphyseal and metaphyseal OO. The medullary and subperiosteal osteoid osteomas are the most common osteoid osteomas of the intraarticular or juxtaarticular locations and are commonly seen within the femoral neck, hands, and feet (Figure 4) [19]. Medullary osteoid osteomas demonstrate eccentric sclerosis of mild to moderate degree. Subperiosteal lesions often do not produce reactive sclerosis and may demonstrate minimal erosion or irregular bony resorption of the subjacent bone, or may manifest as soft-tissue lesions in the vicinity of the affected bone (Figure 5) [19].

Osteoid osteoma is the most common cause of painful scoliosis in pediatric populations (approximately 70%) [20]; the lesion is commonly located along the concavity of scoliosis. Spinal osteoid osteomas are mostly located in the posterior column (neural arch). In the spine, OO mainly affects the lumbar spine. The main role of imaging in osteoid osteoma is to help identify and localize the tumor before surgical or percutaneous treatment.

Skeletal scintigraphy is 100% sensitive for detecting OO and is commonly utilized for localizing OO in clinically suspected cases [21]. The classic scintigraphic finding in the appendicular skeleton is the double-density sign (pinhole magnification scintigraphy better displays the sign than high-resolution planar scintigraphy), characterized by a central area of intense radiotracer uptake related to nidus, surrounded by a larger area of less-intense radiotracer uptake (suggesting the host response) [22]. The role of radiography and scintigraphy is to guide the MR or CT examination to the area of interest.

Dynamic contrast-enhanced CT demonstrates an intense (similar to adjacent arteries) and rapid enhancement of nidus in the arterial phase (attenuation difference of >40 HU) with a persistent venous phase enhancement that, eventually, gradually returns to the baseline attenuation [19]. A tumor nidus demonstrates 18FFDG–avid glucose metabolic activity, whereas surrounding sclerosis does not.

Most tumors are hypo- to isointense on T1-weighted images and have a variable signal intensity on T2-weighted images, which may depend on the tumor stage, its vascularity, and internal matrix mineralization. On MR imaging, the intense bone marrow edema adjacent to the nidus is critical for detection and localization of the lesion. The nidus itself may not be readily seen on MR imaging, depending on its size and the degree of mineralization. Adjacent soft-tissue edema and periosteal reaction are often noted. Reactive changes in the surrounding tissue may mask an intra-articular lesion; although, in a child with joint pain and synovitis, joint effusion, and adjacent marrow edema on MR imaging, OO should be strongly considered. If the diagnosis of OO is still not confirmed, a thin section bone algorithm CT is recommended to demonstrate and precisely localize the nidus. Dynamic, contrast-enhanced MR imaging is recommended for OO and has the sensitivity equal to or greater than that of thin-section CT [23,24]. OO should be suspected if MR reveals edema in the posterior elements, extending to involve the vertebral body with relative sparing of the intervertebral disk space [25].

Osteoid osteomas are stage two lesions and may involute spontaneously over several years. Symptoms can sometimes be managed medically with salicylates or nonsteroidal anti-inflammatory drugs. Surgical excision is an option for patients who cannot tolerate symptoms. En bloc excision of the tumor reduces the risk of local recurrence compared with less aggressive methods, but it poses a risk of subsequent fracture. In weight-bearing bones, en bloc excision may need to be augmented with bone grafting or internal fixation. The radiologist may aid the operating surgeon by the image-guided marking of the lesion with methylene blue before surgical excision, as the lesion may not be visible under a thickened cortex at the time of surgery. The radiologist should also mention the distance of the lesion from a palpable landmark in the CT report [26].

Percutaneous CT guided ablation is now considered a first-line treatment for osteoid osteoma [27]. The nidus is approached using a bone biopsy needle. This technique was first introduced by Rosenthal et al. [28]. Radiofrequency ablation is safe and effective in treating osteoid osteoma [29]. During the ablation, general, spinal or propofol-induced anesthesia is required as local anesthesia may not be sufficient to achieve pain control, particularly during entry into the nidus [30,31]. The average procedure time is 90 min, and the patient may need to be observed for 3–24 h after the procedure [32]. Relative contraindications of radiofrequency ablation include hand lesions and spinal lesions less than 1 cm away from vital structures such as the cord and nerves. A safe and effective CT-guided RF ablation of appendicular and spinal osteoid osteomas can be achieved using a targeted navigational bipolar electrode system [33]. A persistent lucency at the ablated site or increasing arterial enhancement on the follow-up CT and persistent marrow edema on subsequent post-ablation MRI should raise suspicion of residual or recurrent tumors. Magnetic resonance-guided focused ultrasound (MRgFUS) is a novel technique to treat intra-articular osteoid osteoma. Management of intra-articular osteoid osteoma is very challenging due to their local aggressiveness and position. Utilizing MRgFUS, the interventional radiologist can ablate the lesion inside the body without touching the patient. According to the results published by Arrigoni et al., all 14 patients treated with MRgFUS showed significant improvement in pain of about 68% at 6 months and 90% at 12 months [34]. Imaging signs to prove success of CT-guided RFA and MRgFUS include disappearance of bone marrow edema around the lesion, reduction in perilesional synovial reaction, and remodeling of the bone and ring sign, which is identified as a central hypointense area surrounded by peripheral hyperintense area (which is the granulation tissue between the necrotic and viable bone) (Figure 6) [35].

Other minimally invasive techniques include cryoablation, ethanol injection, interstitial laser photocoagulation, and arthroscopic excision.

The differential diagnosis for OO in the long bones includes stress fractures and osteomyelitis (Brodie abscess). Dynamic contrast-enhanced CT helps differentiate osteoid osteoma from bone cysts and chronic osteomyelitis, specifically Brodie abscess. The marked early enhancement for osteoid osteomas is in contradistinction to Brodie abscesses and bone cysts, which are avascular. Toxic and inflammatory synovitis can be considered in the differential diagnosis for intra-articular lesions. Compared to the rounded or ovoid lucency and double-density sign of OO, a stress fracture presents as a linear infraction in the center of the cortical thickening and shows linear radiotracer uptake. To differentiate intracortical abscess from osteoid osteoma, it is vital to correlate imaging findings (diffuse enhancement of nidus versus rim enhancement of an abscess) with clinical and laboratory findings (such as blood cultures or presence of adjacent orthopedic hardware to rule out infection).

## 6. Osteoblastoma

Osteoblastoma, a benign primary osteogenic tumor of bone, accounts for approximately 1% of all primary bone tumors [36] and 3% of all benign bone tumors. It is commonly seen in young (10–30 years) males (M: F = 2:1) [37,38]. Approximately 30–40% of all osteoblastomas occur in the spine [39].

Though osteoid osteoma and osteoblastoma are histologically similar, they differ by their clinical manifestations and natural evolution. The most crucial criterion is size: osteoblastomas are usually larger than 2 cm [40]. Compared to the worsening night pain of osteoid osteoma, osteoblastoma is either asymptomatic or may cause dull pain. Because pain caused by an osteoblastoma is not related to the inflammatory reaction (compared to osteoid osteomas), analgesics usually fail in achieving effective and long-lasting pain relief. Additionally, instead of involuting, OB tends to be locally aggressive. In a typical osteoblastoma, the bone trabeculae are broader and less densely packed, and less coherent than those in osteoid osteoma. OB usually lacks the halo of sclerotic bone, which is characteristic of osteoid osteoma.

The typical locations in the skeleton are the spine (originating in the neural arch and extending into the vertebral body) and long bones (eccentrically located in diaphysis or metaphysis). Children with vertebral involvement may present with torticollis or scoliosis with or without back pain. Benign osteoblastoma has a distinct predilection for the spine, with 36% of all osteoblastoma and 10% of all osteoid osteomas occurring in the spine and sacrum [41]. In the long bones, the femur and tibia are most frequently involved with a round or oval lucent proximal diaphyseal lesion with matrix mineralization. The third most common tumor location is the bone of the hands and feet (talus most involved).

Three different radiographic presentations of osteoblastoma (OB) are known. It may present as a “giant osteoid osteoma.” A spinal osteoblastoma may appear as an expansile lesion with a mineralized matrix and a narrow zone of transition (Figure 7). It may have a more aggressive appearance in long bones with an expansile lesion, with associated partial cortical destruction and soft tissue infiltration.

Radionuclide bone scintigraphy invariably demonstrates an intense focal increase of activity and can be of great assistance in demonstrating and localizing disease to the spine [42].

CT can better demonstrate the mineralization and the lesion margin. In complex bony structures, such as in the spine or pelvis, CT demonstrates the extent of the lesion and helps differentiate between bone and soft tissue components.

MR imaging may show reactive marrow edema of the surrounding bone, usually less than with OO, with a variable degree of increased T2 signal depending on matrix mineralization and the cellularity of the lesion. MR imaging helps to determine the extension of the spinal lesions into the neural foramina and spinal canal. A secondary aneurysmal bone cyst may occur in 16% of OBs (Figure 8) [8]. T2 signal depends on the degree of tumor matrix mineralization. Osteoblastoma demonstrates post contrast enhancement, given the vascular nature of the lesion. Spinal osteoblastoma can be treated with curettage and bone grafting. Preoperative embolization has been found to be helpful [43]. The recurrence rate in surgically treated OBs is approximately 15–20% [39].

Stage two and stage three osteoblastoma differ in their responses to treatment. For stage two lesions that are well contained within the bone, treatment with curettage and grafting suffices. However, stage three lesions need surgical excision of the tumor to prevent recurrence [44]. Some patients will be cured following incomplete excision, but this must be followed up aggressively for local recurrence. The role of radiation therapy remains controversial. Radiation may be indicated in cases of spinal lesions with an extensive soft-tissue component and epidural spread or following local recurrence.

Osteoblastoma can be treated either by surgery or by minimally invasive percutaneous image-guided treatments (such as radiofrequency ablation or cryoablation). Percutaneous image-guided cryoablation represents an effective therapeutic option for patients presenting with painful osteoblastoma [45].

## 7. Conclusions

By becoming familiar with the clinical and radiological characteristics of osteocytic bone tumors in children, physicians will be able to accurately establish the correct diagnosis without the need for biopsy in most cases. With an understanding of the natural history of the tumor and its stage at presentation, appropriate treatment recommendations can be made, thereby preventing both overtreatment and undertreatment of these tumors. Enostoses are “do not touch” lesions. Osteoid osteomas are benign, painful bone tumors with a predilection for the lower extremities in young male patients. Increased recognition of osteoid osteomas and their various imaging manifestations should facilitate more rapid diagnosis and referral for treatment. An osteoblastoma is the primary osseous lesion of the spine that needs special attention by a radiologist, as these patients often present with nonspecific symptoms—cross-sectional imaging aids in planning a biopsy, staging of the tumor, and surgical planning. The following Table 1 highlights key features to differentiate benign osteocytic tumors in children.

## Figures and Tables

**Figure 1 jcm-10-02823-f001:**
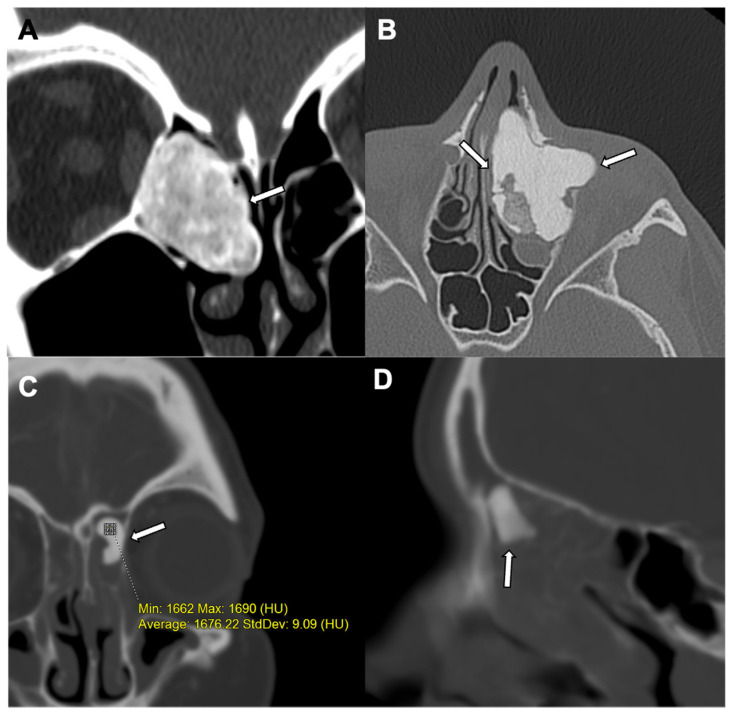
Osteoma: A juxtacortical circumscribed homogeneous sclerotic lesion (white arrow) in the right anterior ethmoid sinus is revealed in a coronal CT of the facial bone window, suggestive of an osteoma in a 6-year-old male (**A**). A 16-year-old female with compact and trabecular sclerotic polypoidal osteoma (white arrow) is seen in the left anterior ethmoid sinus with associated mild ethmoidal sinusitis in axial CT of the sinus bone window (**B**). An osteoma causing mild mass effect on the adjacent left inferomedial orbital wall. Another well-defined sclerotic lesion in the left frontoethmoidal recess (white arrows) is seen on coronal and sagittal CTs of the sinus bone window (**C**,**D**) with moderate pan sinusitis in a 7-year-old male. Note is made of very high HU (Hounsfield Unit) values.

**Figure 2 jcm-10-02823-f002:**
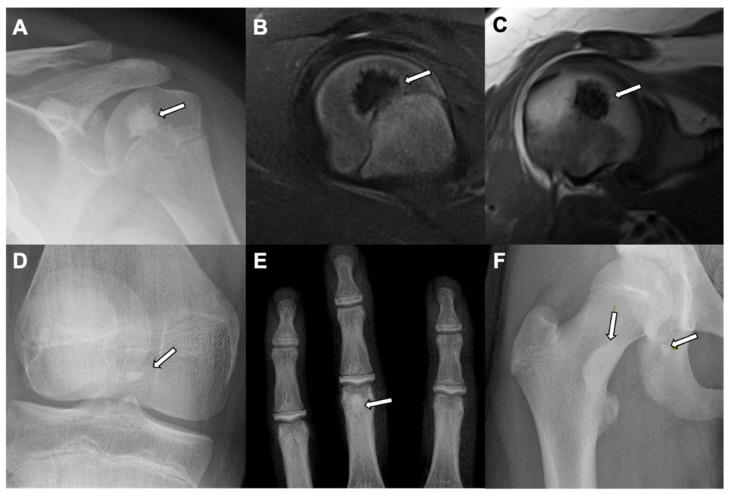
Bone island: A left shoulder X-ray in a skeletally immature young patient with skeletally immature bones show: (**A**) homogeneous intramedullary sclerotic focus with spiculated margin in the humeral head (white arrow). An MRI of the left shoulder of the same patient showed diffuse T1 and T2 hypointensity with spiculated margin of the lesion (white arrows) seen on sagittal T2 fat saturated sequence (**B**), and coronal proton density sequence (**C**) suggests a benign sclerotic lesion-bone island. Further examples of bone islands are seen as homogeneous intramedullary sclerotic focus (white arrows) in the tibial epiphysis in the intercondylar region of the right knee X-ray (anteroposterior view) (**D**), at the distal end of the proximal phalanx of the middle finger of the left hand X-ray (anteroposterior view) (**E**), and in the right medial femoral neck and right ischium of the right hip X-ray (anteroposterior view) (**F**) in young patients with skeletally immature bones.

**Figure 3 jcm-10-02823-f003:**
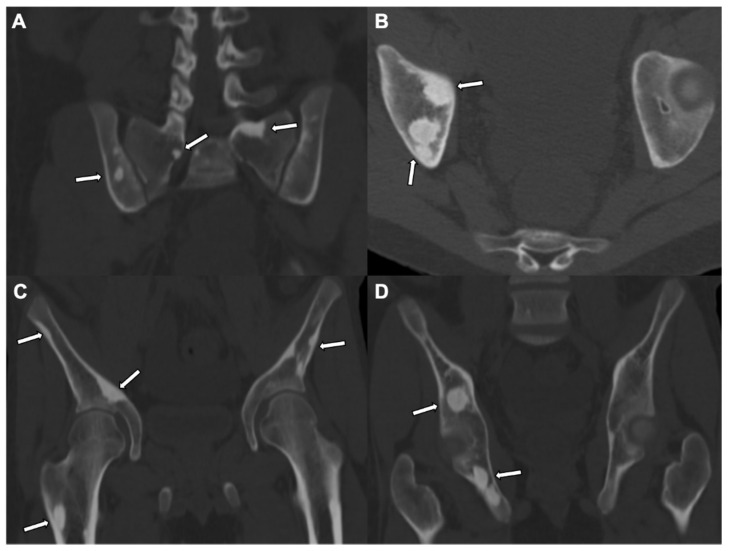
Osteopoikilosis: A CT pelvis, in coronal (**A**,**C**,**D**) and axial (**B**) views, of bone windows in a 17-year-old patient shows scattered, rounded, or well defined intramedullary sclerotic foci in bilateral iliac bones, bilateral sacral promontories, the right superior pubic ramus, ischium, and femur subtrochanteric proximal femoral shaft (white arrows), suggesting benign hereditary etiology such as osteopoikilosis.

**Figure 4 jcm-10-02823-f004:**
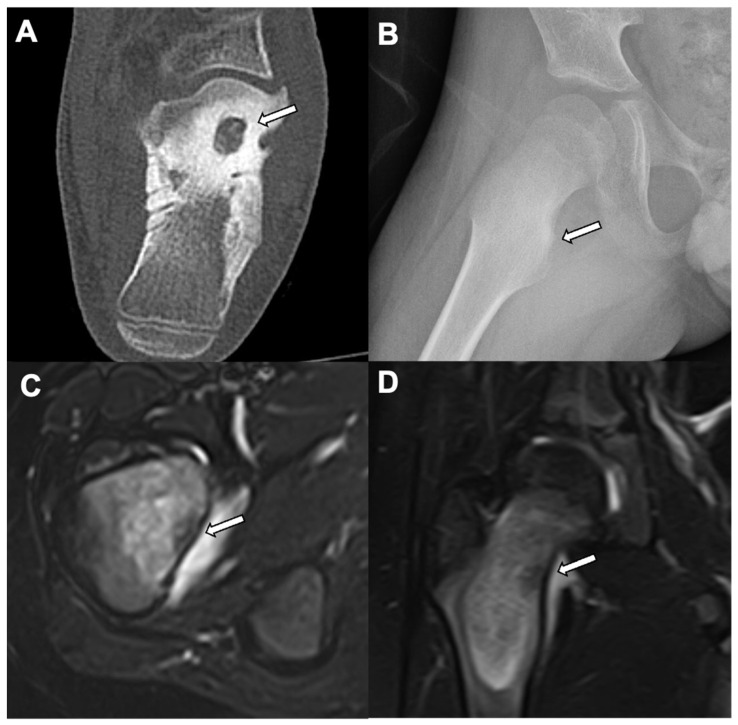
Osteoid osteoma: A left foot CT of the bone window (axial view) shows a central well defined lytic lesion along the anterosuperior calcaneum with diffuse surrounding sclerosis (white arrow) in a 10-year-old male (**A**). Another 12-year-old patient with nighttime right hip pain showed focal ill-defined sclerosis along the right medial femoral neck cortex (white arrow) (**B**). This patient’s MRI shows focal T2 hypointensity in the right medial femoral neck cortex with surrounding bone marrow edema (white arrows) of the femoral neck in T2 fat saturation (axial (**C**) and coronal (**D**) views).

**Figure 5 jcm-10-02823-f005:**
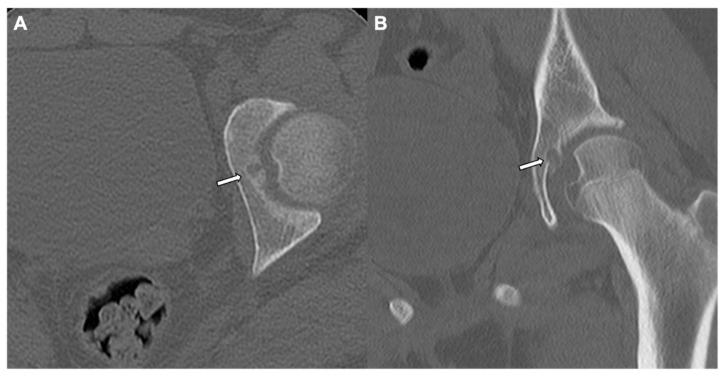
Subperiosteal osteoid Osteoma: A left hip CT bone window (axial view (**A**) and coronal view (**B**)) in a 14-year-old male shows well defined lucency with focal cortical truncation involving the left superomedial acetabular surface without surrounding sclerosis (white arrows) and suggests an intraarticular subperiosteal osteoid osteoma.

**Figure 6 jcm-10-02823-f006:**
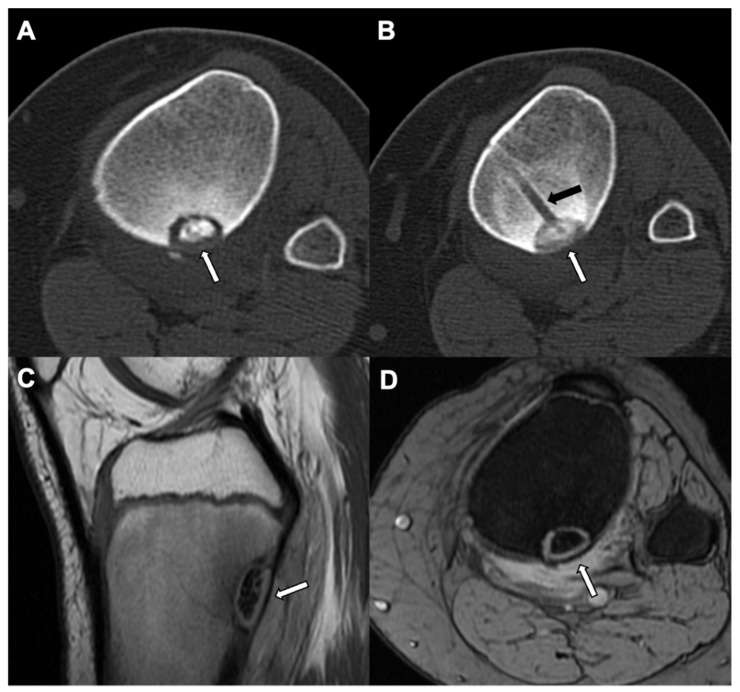
Osteoid osteoma with RF ablation: Left leg CT bone window axial view (**A**) in a 13-year-old patient shows well defined sclerosis with surrounding lytic margin along the posterior tibial cortex (white arrow), suggesting an osteoid osteoma. Status post RF ablation of the same lesion (**B**) showed developing minimal scattered sclerosis in lytic component (white arrow) with an ablation track (black arrow). MRI left leg sagittal T1 (**C**) and axial proton density image (**D**) of the same patient shows diffuse ring sign which is identified as central hypointense area surrounded by peripheral hyperintense area, suggesting favorable response to ablation (white arrows).

**Figure 7 jcm-10-02823-f007:**
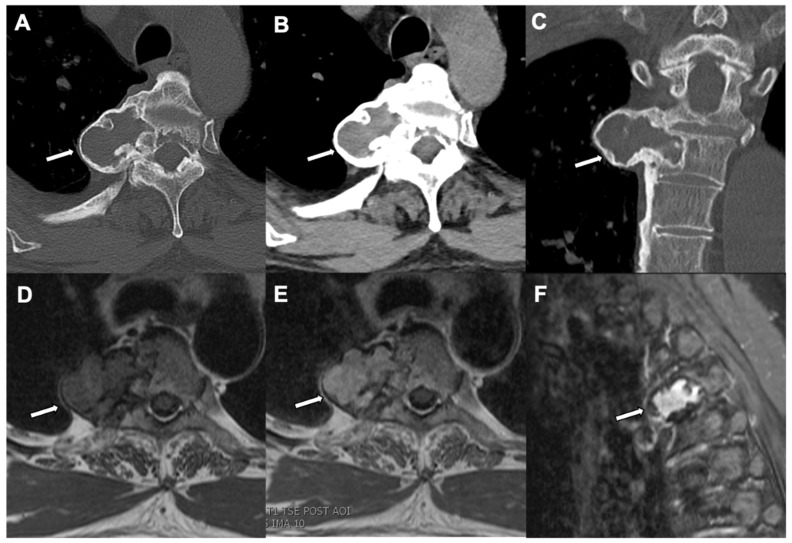
Osteoblastoma: A CT Chest bone window (axial view) (**A**), a soft tissue window (**B**), and CT chest bone window (coronal view) (**C**) in an 18-year-old patient show an expansile lytic lesion (white arrows) with a mineralized matrix and a narrow zone of transition arising from the right pedicle of an upper thoracic vertebra and extending into the vertebral body. An MRI of the chest of the same patient shows a T1 hypointense signal (**D**), a T2 hyperintense signal (**E**), and diffuse enhancement of the vertebral lesion (**F**) in an axial T1-weighted sequence, axial T2-weighted sequence, and sagittal fat saturated T1-weighted postcontrast sequence, respectively.

**Figure 8 jcm-10-02823-f008:**
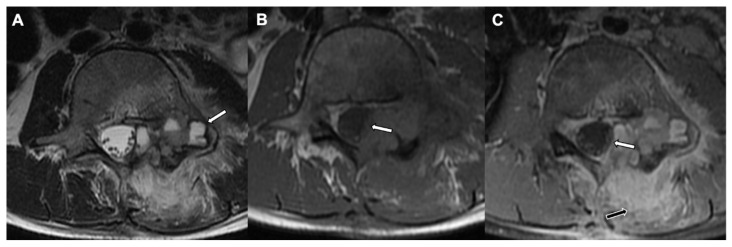
Osteoblastoma with secondary aneurysmal bone cyst formation in an 11-year-old female: MRI of the lumbar spine shows a well-defined expansile lesion involving left lamina and pedicle with fluid—fluid levels (white arrow) on the axial T2-weighted sequence (**A**). The lesion encroaching in the spinal canal causing mild abutment of the traversing nerve fibers is seen on the axial T1-weighted sequence (**B**) and the axial T1-weighted postcontrast image (**C**). A note is also made of associated diffuse moderate to advanced left posterior paraspinal muscle strain with enhancement (black arrow) and mild left psoas muscle edema.

**Table 1 jcm-10-02823-t001:** Important features to differentiate pediatric benign osteocytic bone tumors.

	Osteoma	Enostosis	Osteoid Osteoma	Osteoblastoma
Age	Between 3rd and 5th decades	No age predilection	Between 5 and 25 years of age	Between 10 and 30 years of age
Presentation	Usually asymptomatic; may present with headache and sinusitis	Most commonly incidental on imaging	Worsening night pain, relieved by non-steroidal anti-inflammatory drugs	Dull pain, not relieved by non-steroidal anti-inflammatory drugs
Lesion Location	Paranasal sinuses, mainly frontal and ethmoidal	Axial skeleton (spine, pelvis, and ribs); long bones	Long bones, mainly femur	Spine and long bones
Radiographic/CT features	Juxtacortical, well-circumscribed homogenous sclerotic lesion	Homogenous intra-medullary sclerotic focus with spiculated margins	Small (less than 2 cm) cortical lucency with extensive surrounding sclerosis	Expansile large (more than 2 cm) lucent lesion with matrix mineralization
Treatment	None if asymptomatic; excision if complications related to mass effect	None	Percutaneous CT guided radiofrequency ablation	Surgery or percutaneous CT guided ablation

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
