# Peer review of "Imaging Review of Pediatric Benign Osteocytic Tumors and Latest Updates on Management"

_jcm, 2021, doi:10.3390/jcm10132823_

Round 1

Reviewer 1 Report

I suggest to add the word "osteoma/enostosis" at the beginning of the abstract, as they are related conditions.

Musculoskeletal tumor society should be with capitalized first letters.

When you describe osteoma, you shouldn't use the word osteoid, because this tumor does not produce osteoid.

Is most found-> is most often found

I would specify that mature osteoma can include trabecular bone often with marrow

It also causing mild mass effect -> it also causes

In Fig. 6 replace both suggests with suggesting.

Author Response

I suggest to add the word "osteoma/enostosis" at the beginning of the abstract, as they are related conditions.: Changed as requested

Musculoskeletal tumor society should be with capitalized first letters: Changed as requested

When you describe osteoma, you shouldn't use the word osteoid, because this tumor does not produce osteoid: Changed as requested

Is most found-> is most often found: Changed as requested

I would specify that mature osteoma can include trabecular bone often with marrow: Changed as requested

It also causing mild mass effect -> it also causes: Changed as requested

In Fig. 6 replace both suggests with suggesting.: Changed as requested

Thanks for all the suggestions. 

Reviewer 2 Report

The manuscript was well-written, but I feel it is not sufficient as a review article for pediatric osteolytic tumors.

Comments:

  1. How the authors defined “benign osteolytic tumors”? Please show textbook, classification or previous article.
  2. Related to above comment, other common osteolytic tumors (tumor-like lesion) such as fibrous dysplasia, Langerhans cell histiocytosis and simple bone cyst was not included in this review article. Please explain why did the authors excluded those lesions.
  3. There are several cases in the manuscript shown in the Figures. As this manuscript discusses “pediatric” tumors, the cases should be pediatric cases. Please add age of the patients.
  4. In page 3, differential diagnosis of bone islands (osteopoikilosis?) was described. However, prostate and breast cancers were raised as differential diagnoses, which is not likely to occur in “pediatric“ cases. Please consider revising the differential diagnoses.
  5. As a characteristic finding for bone island, “salt-and-pepper noise sign” was described, however, no image was shown. It would be helpful if any characteristic image was shown.
  6. Incidence of “giant osteoid osteoma” in children should be shown.
  7. It was described that secondary ABC may happen in 16% of OBs in page 11. Please show the source of the data and whether there is a difference in incidence between adult and pediatric cases.
  8. It is not common to apply ablation for osteoblastoma. Is that specific for pediatric patients to preserve normal tissues? Please show other references than ref#46 and explain in detail.
  9. It would be helpful to show table or flow chart to distinguish osteolytic tumors in pediatric patients.

Author Response

  1. How the authors defined “benign osteolytic tumors”? Please show textbook, classification, or previous article.
  2. Related to above comment, other common osteolytic tumors (tumor-like lesion) such as fibrous dysplasia, Langerhans cell histiocytosis and simple bone cyst was not included in this review article. Please explain why the authors excluded those lesions.

Reply: The abstract was selected by guest editors as a planned paper on special issue “Recent research in diagnosis and treatment of benign osteocytic tumors (Osteoma, Osteoid Osteoma and Osteoblastoma). Here is the link to the special issue:

https://www.mdpi.com/journal/jcm/special_issues/Benign_Osteocytic_Tumours

That is why radiolucent/osteolytic tumors such as fibrous dysplasia, Langerhans cell histiocytosis and simple bone cyst were not described in the manuscript.

As a principal author of this manuscript, I was not sure whether I should use the term osteocytic or osteogenic (preferred in latest WHO classification of bone tumors), but as the title of special issue was termed as “osteocytic” and my abstract was accepted as a planned paper by world expert guest editors, I choose to use the term osteocytic.

Do you want me to change the term to osteogenic?

Following are the references suggesting the classification of benign osteogenic tumors/osteoid matrix containing tumors:

[1-3]

[1]          K. Motamedi and L. L. Seeger, "Benign bone tumors," Radiol Clin North Am, vol. 49, no. 6, pp. 1115-34, v, Nov 2011, doi: 10.1016/j.rcl.2011.07.002.

[2]          S. Ahlawat and L. M. Fayad, "Revisiting the WHO classification system of bone tumours: emphasis on advanced magnetic resonance imaging sequences. Part 2," Pol J Radiol, vol. 85, pp. e409-e419, 2020, doi: 10.5114/pjr.2020.98686.

[3]          J. H. Choi and J. Y. Ro, "The 2020 WHO Classification of Tumors of Bone: An Updated Review," Adv Anat Pathol, vol. 28, no. 3, pp. 119-138, May 1 2021, doi: 10.1097/PAP.0000000000000293.

I sincerely apologize for the confusion. Please advise me

  1. There are several cases in the manuscript shown in the Figures. As this manuscript discusses “pediatric” tumors, the cases should be pediatric cases. Please add age of the patients. :

Reply: Yes, all patients were pediatric patients. I added the age of the patients in figure description with underlined changes.

  1. In page 3, differential diagnosis of bone islands (osteopoikilosis?) was described. However, prostate and breast cancers were raised as differential diagnoses, which is not likely to occur in “pediatric“ cases. Please consider revising the differential diagnoses.

Reply: Removed metastases line and replaced with multiple sclerotic bone lesions seen in tuberous sclerosis complex.

  1. As a characteristic finding for bone island, “salt-and-pepper noise sign” was described, however, no image was shown. It would be helpful if any characteristic image was shown.

Reply: Unfortunately, I do not have the image, so I removed that information from the manuscript, removed the citation and rearranged all the remainder of citations. Thanks.

  1. Incidence of “giant osteoid osteoma” in children should be shown.

             Reply: I could not find the incidence of giant osteoid osteoma in children in literature. I apologize.

  1. It was described that secondary ABC may happen in 16% of OBs in page 11. Please show the source of the data and whether there is a difference in incidence between adult and pediatric cases.

Reply: Added reference for the source. I could not find the difference in incidence between adult and pediatric cases.

  1. Motamedi and L. L. Seeger, "Benign bone tumors," Radiol Clin North Am, vol. 49, no. 6, pp. 1115-34, v, Nov 2011, doi: 10.1016/j.rcl.2011.07.002.
  2. It is not common to apply ablation for osteoblastoma. Is that specific for pediatric patients to preserve normal tissues? Please show other references than ref#46 and explain in detail.

Reply: Replaced the existing reference with AJR reference.

  1. L. Cazzato et al., "Percutaneous Image-Guided Cryoablation of Osteoblastoma," AJR Am J Roentgenol, vol. 213, no. 5, pp. 1157-1162, Nov 2019, doi: 10.2214/AJR.19.21390.
  2. It would be helpful to show table or flow chart to distinguish osteolytic tumors in pediatric patients.

Reply: Added the following chart on distinguishing osteocytic tumors in pediatric patients. Thanks.

Osteoma

Enostosis

Osteoid Osteoma

Osteoblastoma

Age

Between 3rd and 5th decades

No age predilection

Between 5 and 25 years of age

Between 10 and 30 years of age

Presentation

Usually asymptomatic; may present with headache and sinusitis

Most commonly incidental on imaging

Worsening night pain, relieved by non-steroidal anti-inflammatory drugs

Dull pain, not relieved by non-steroidal anti-inflammatory drugs

Lesion Location

Paranasal sinuses, mainly frontal and ethmoidal

Axial skeleton (spine, pelvis, and ribs); long bones

Long bones, mainly femur

Spine and long bones

Radiographic/CT features

Juxtacortical, well-circumscribed homogenous sclerotic lesion

Homogenous intra-medullary sclerotic focus with spiculated margins

Small (less than 2 cm) cortical lucency with extensive surrounding sclerosis

Expansile large (more than 2 cm) lucent lesion with matrix mineralization

Treatment

None if asymptomatic; Excision if complications related to mass effect

None

Percutaneous CT guided radiofrequency ablation

Surgery or percutaneous CT guided ablation

Round 2

Reviewer 2 Report

The authors sufficiently revised the manuscript in response to my comments. 

Regarding the definition of "osteolytic tumor", I would like to ask the guest editor to judge. 

Author Response

The authors sufficiently revised the manuscript in response to my comments. 

Regarding the definition of "osteolytic tumor", I would like to ask the guest editor to judge. 

Reply: Yes I agree. Thanks.